# Bovine Rumen Microbiome: Impact of DNA Extraction Methods and Comparison of Non-Invasive Sampling Sites

**Alexander C. Mott** [1,*] , **Dominik Schneider** [2] , **Martin Hünerberg** [1] , **Jürgen Hummel** [1] and **Jens Tetens** [1,3]

1 Department of Animal Science, Georg-August-University of Göttingen, 37077 Goettingen, Germany; martin.huenerberg@uni-goettingen.de (M.H.); jhummel@gwdg.de (J.H.); jens.tetens@uni-goettingen.de (J.T.)
2 Department of Genomic and Applied Microbiology, Georg-August-University of Göttingen, 37077 Goettingen, Germany; dschnei1@gwdg.de
3 Center for Integrated Breeding Research, Georg-August-University of Göttingen, 37075 Goettingen, Germany
* Correspondence: alexandercharles.mott@uni-goettingen.de; Tel.: +49-551-3923268

**Abstract:** With increasing global demand for animal protein, it is very important to investigate the impact of the bovine rumen microbiome on its functional traits. In order to acquire accurate and reproducible data for this type of study, it is important to understand what factors can affect the results of microbial community analysis, and where biases can occur. This study shows the impact of different DNA extraction methods on microbial community composition. Five DNA extraction methods were used on a ruminal sample. These experiments expose a high level of variability between extraction methods in relation to the microbial communities observed. As direct access to the rumen is required, we also investigated possible alternative sampling sites that could be utilised as non-invasive indicators of the bovine rumen microbiome. Therefore, oral swabs and faecal samples were taken in addition to ruminal samples, and DNA was extracted using a single method, reducing bias, and analysed. This is a small pilot study intending to reinforce the need for a universally used methodology for rumen microbiome analysis. Although alternative sampling points can indicate some of the communities present in the rumen, this must be approached cautiously, as there are limits to the depth of community analysis possible without direct rumen sampling.

**Keywords:** 16S rRNA; 18S rRNA; DNA extraction; microbiome; rumen

## 1. Introduction

As the world population increases, so too will the global demand for animal protein, with estimates suggesting that global meat and milk production will have to increase by more than 60% by 2050 [1]. Feed costs typically accounts for 60–70% of total expenditure in beef and dairy production [2], whilst requiring substantial land mass for feed production [3].

Understanding the underlying mechanisms that influence feed efficiency, particularly with respect to the involvement of the rumen microbiome, can aid in increasing the efficiency and sustainability of ruminant production [4]. Therefore, there is a need to understand how the utilisation of feed by the rumen microbiome occurs and enhance the nutrient availability to the host. A better understanding of compositional changes in the rumen microbiome can potentially lead to improvements in production efficiency, which is central to ensuring food security.

By harbouring a complex consortia of microorganisms, such as anaerobic bacteria, protozoa, fungi, and methanogenic archaea, the rumen is uniquely adapted to break down complex carbohydrates, such as hemicellulose and cellulose [5]. The end products of microbial rumen fermentation are then absorbed by the host animal and utilised as substrate for metabolic and productive purposes [5]. Ruminant livestock production has been estimated to be responsible for approximately 14% of anthropogenic methane, a potent greenhouse gas, through the activity of rumen methanogens [6]. The released methane, produced by rumen methanogens, is a major problem for the environment, but losses of

enteric methane also accounts for up to 12% of gross energy intake and therefore represents a significant loss of energy to the host [7,8]. This makes the rumen microbiome and its ability to efficiently and effectively process feedstuffs highly important to future global food security. Attempts to manipulate the rumen microbiome to benefit global agricultural challenges have been ongoing for decades with limited success, mostly due to the lack of a detailed understanding of the microbiome [9] and our limited ability to culture most of these microbes outside the rumen [10]. Gathering a deeper understanding of the rumen microbiome will ultimately help to solve current and future ruminant livestock challenges.

In the last decades, molecular biology tools have been utilised to determine changes and functions within microbial consortia. Using metagenomic techniques allows for high-resolution observation of potential changes in rumen microbial populations with respect to composition and level of metabolic activity. The value of the information generated by these innovative tools is limited by a lack of standardised methods for collecting, processing and analysing the samples, resulting in data that is often difficult to directly compare across studies [11–16].

It is crucial that future strategies are designed in such a way that data can be extrapolated and combined to give a deeper understanding of the microbiome as a whole. Using such comparable study methods would allow for the manipulation of experimental conditions within the rumen in such a way as to remove biases and environmental impacts that are not in the scope of the studies. This would enhance future experimental design strategies that can help to improve efficiency and reduce the environmental impact of ruminants as a whole.

Functional traits, such as residual feed intake (RFI), are monitored on many farms, and are used as an indicator of feeding efficiency when selecting cattle for further breeding [17]. Several studies have indicated that changes in feed digestion, changes in metabolism and body composition can be linked to a divergent RFI [17,18]. Accurate longitudinal microbiome studies would therefore enhance the ability of farmers to link phenotypes to changes in the microbiome, and can even lead to the selection of specific phenotypes through the adjustment of the microbiome in conjunction with and enhancing an established breeding programme.

In order to perform these types of analysis, farmers must have a simple and easy way in which to sample a large number of animals. The current sampling methods for the collection of rumen samples are invasive and involve a high level of skill to perform adequately. These being either the insertion of a gastric sampling tube into the rumen (rumenocentisis, the collection of rumen fluid by percutaneous needle aspiration), or via rumen fistulation of the animal, all of which have their limitations. The gastric sample tube can be susceptible to saliva contamination and can be difficult to know the exact location in which the tube is situated [16,19,20], and both rumenocentisis and fistulation can run the risk of external contamination of the sample, as well as needing close monitoring of the animal to avoid the occurrence of infection [21,22]. As such, a non-invasive sampling method that can be conducted in many animals would aid in this approach. As hologenomic selection approaches require a microbial similarity matrix based on measures with high repeatability [23,24], and oral sampling is depending on the exact time of rumination, the use of faecal sampling can be suitable to show explained variances under highly repeatable samplings.

It is also important to understand that, when investigating the rumen microbiome, the method in which the studies are performed can also have an impact and lead to bias in the results that are produced. For instance, different DNA extraction protocols can lead to different results, with respect to the microbial diversity and community structure, which makes the final comparison of studies very difficult, as one cannot curate the data for all these biasing factors when comparing studies [11,12,14].

There are also a number of other variables that can drive bias in microbiome studies. These include primer selection, PCR component choice and cycling protocol, as well as the sequence methodology used [25]. Additionally, bioinformatic data processing can also drive

bias, although these can be accounted for if the raw data is available to re-interpret [26–28]. This study examines several different extraction methods to highlight specific pitfalls when determining the diversity of the rumen bacterial community. Furthermore, we investigated non-invasive sampling points (oral and faecal) as proxy for the rumen microbiome of Holstein cattle, and bring together the results of previous studies.

## 2. Materials and Methods

### 2.1. Ethical Statement

The following study was carried out in accordance with the EU Directive 2010/63/EU for animals used for scientific purposes, under the current animal licence 18A269, and all sampling was carried out in accordance with the German Animal Welfare Act approved by the LAVES (Lower Saxony State Office for Consumer Protection and Food Safety, Germany).

### 2.2. Animals and Sample Collection

Two ruminally cannulated, non-lactating Holstein Friesian cows were utilised in this pilot study. The cows were housed in a free stall with ad libitum access to mixed grass hay. Twice per day (730 and 1400 h), cows were offered approx. 200 g of pelleted concentrate containing mainly barley grain, canola meal and dry beet pulp. Both cows had free access to water and a trace mineral and salt block.

Samples were taken at two time-points on the same day, which were then combined per extraction to reduce diurnal feeding effects. Samples were taken 30 min before feeding in the morning and 6 h later. Three sample types were collected; buccal fluid, composite rumen samples and faeces. The buccal fluid collection was performed as follows: the animals were incentivised to bring the bolus forward with feed being placed just out of reach. The sterile salivette® collection cotton swabs (Sarstedt, Nuembrecht, Germany) were then placed in the mouths of the cows using sterile forceps and rubbed for 1 min within the mouth (on the walls). A total of 1 mL of PBS solution was added to the swab, which was vigorously agitated at 50 °C in an orbital shaker (Sartorius, Goettingen, Germany) for 1 h. The swabs were then centrifuged at $10,000 \times g$ to remove all collected sample from the swab and the bacteria was eluted and collected. This was then used in the further extraction method. This method allowed for a reasonable collection of saliva and oral microorganisms similar to previously published methods [16,29]. Composite rumen samples (approx. 750 g) were obtained from three sites (reticulum, dorsal and ventral sac) within the rumen of each cow through the cannula. Immediately after collection, the samples were separated into three fractions. Firstly, the complete sample, then by manually pressing the remains of each sample through two layers of sterile medical gauze collecting the pressed solid fraction, and also collecting the filtered liquid sample in a sterile container. The faecal samples were collected by stimulating rectal activity in order to generate fresh material. Faecal samples were taken approximately 60 cm deep inside the rectum using sterile gloves. Approximately 500 g of faecal matter was collected, from which a smaller sample (50 g) was taken for this study. These samples were then taken for further evaluation and placed in a sterile container. To avoid any process time bias all samples were stored at −80 °C directly after collection.

### 2.3. DNA Extraction Methodologies

Three commercial extraction protocols with and without bead-beating steps and two published methodologies were tested for their efficacy at extracting high quality DNA. Extraction methods (as shown in Table 1) were performed as stated in the kit manuals (Qiagen, Hilden, Germany/Omni-International, Kennesaw, GA, USA) with the addition of a bead-beating step (BB) where stated. This was performed at 6 m/s using a Bead Ruptor (Omni international, Germany) for $2 \times 30$ s, using a mix of 400 mg 0.1 mm and 600 mg 0.5 mm ceramic beads. The repeat bead beater and column (RBBC) method was adapted from Yu and Morrison (2004), and the phenol extraction (PC) method was adapted from Griffiths et al. (2000). As DNA extracted from rumen samples can easily be contaminated

with components (e.g., humic compounds) that can inhibit PCR reactions, each extracted DNA sample was analysed using a Tecan Infinite 200 Pro and Nanoquant plate (Tecan, Crailsheim, Germany). Here the absorbance at 260 nm and 280 nm (A260/280) of each extracted sample was analysed to determine DNA quality and quantity for downstream sequencing use. All methods were performed in triplicate, with mean average shown in the final Table 1. Integrity was determined by agarose (1% *w*/*v*) gel electrophoresis and run for 2.5 h at 80 V using 1 Kb Plus DNA Ladder (Thermo Fisher Scientific, Osterode am Harz, Germany) as molecular weight markers, post-staining with HDGreen DNA gel stain (Intas, Goettingen, Germany) and illumination under UV light.

**Table 1.** Extraction methodology overview.

| Extraction Method | Abbreviation | Comments | Reference |
|---|---|---|---|
| QIAamp Fast DNA Stool Mini Kit | QS | Followed manufacturer's instructions | Qiagen®, Hilden, Germany |
| QIAamp Fast DNA Stool Mini Kit+ Bead Beating | QSB | As above, including bead beating step | Qiagen®, Hilden, Germany |
| QIAamp DNA Microbiome Kit | QM | Followed manufacturer's instructions | Qiagen®, Hilden, Germany |
| QIAamp DNA Microbiome Kit + InhibitEX Buffer | QMI | As above, including use of InhibitEX buffer | Qiagen®, Hilden, Germany |
| Soil DNA Purification Kit | OS | Followed manufacturer's instructions | Omni-International®, Kennesaw, GA, USA |
| Repeat Bead Beater and Column Method | RBBC | As stated in reference | [30] |
| Phenol Chloroform Extraction | PC | As stated in reference | [31] |
| Phenol Chloroform Extraction + Bead Beating | PCB | As above, including bead-beating step | [31] |

*2.4. Amplification and Sequencing of Ribosomal RNA Genes*

Amplification of the 16S rRNA gene sequences was performed by the utilisation of primers targeting the V3-V4 region (D-Bact-0341-b-S-17, TCGTCGGCAGCGTCAGAT GTGTATAAGAGACAGCCTACGGGNGGCWGCAG-; S-D-Bact-0785-a-A-21, GTCTCGTG GGCTCGGAGATGTGTATAAGAGACAGGACTACHVGGGTATCTAATCC) [32]. The 16S rRNA PCR was performed as follows: the reaction (50 μL) contained 10 μL of 5x Phusion GC buffer, and final concentrations of 10 mM MgCl$_2$, 0.25% DMSO, 200 μM of each of the 4 deoxynucleoside triphosphates and 1 U of Phusion DNA Polymerase (Thermo Fisher Scientific, Osterode am Harz, Germany). A total of 25 ng of extracted DNA was used as a template per reaction. The PCR reaction was started by an initial denaturation at 98 °C for 1 min, followed by 25 cycles of denaturation at 98 °C for 45 s, annealing at 55 °C for 45 s and elongation at 72 °C for 30 s. The final elongation was performed at 72 °C for 5 min.

In order to amplify 18S rRNA gene sequences, primers targeting the conserved regions adjoining the 5′ and 3′ region of the V4 rRNA loop (TAReuk454FWD1, TCGTCGGCAGCGT CAGATGTGTATAAGAGACAG-CCAGCASCYGCGGTAATTCC; TAReukREV3, GTCTCG TGGGCTCGGAGATGTGTATAAGAGACAG-ACTTTCGTTCTTGATYRA) [33]. The 18S rRNA gene PCR was performed as follows: the reaction (50 μL) contained 10 μL of 5x Phusion GC buffer, and final concentrations of 10 mM MgCl2, 0.25% DMSO, 200 μM of each of the 4 deoxynucleoside triphosphates and 1 U of Phusion DNA Polymerase (Thermo Fisher Scientific, Osterode am Harz, Germany). A total of 50 ng of extracted DNA was used as a template per reaction. The PCR reaction was started by an initial denaturation at 98 °C for 1 min, followed by 25 cycles of denaturation at 98 °C for 45 s, annealing at 47 °C for 45 s and elongation at 72 °C for 30 s. The final elongation was performed at 72 °C for 5 min.

Amplicons were then purified by using the MagSi-NGS PREP Plus magnetic beads (AMS Biotechnology, Abingdon, UK), with 30 μL bead solution on 25 μL amplicon solution and an elution volume of 30 μL EB buffer. Purified amplicons were sequenced with an

Illumina MiSeq and Nextera XT DNA Library Prep Kit chemistry (Illumina, San Diego, CA, USA), resulting in paired-end reads of 2 × 300 base pairs length.

*2.5. Sequence Processing and Analyses*

Demultiplexing and clipping of adapter sequences from the raw amplicon sequences were performed with the CASAVA software (Illumina). The programme fastp (v0.20.0) [34] was used for quality filtering with a minimum phred score of 20, a minimum length of 50 base pairs, a sliding window size of four bases, read correction by overlap and adapter removal of the Illumina Nextera primers. Paired-end reads were merged with the paired-end read merger (PEAR v.0.9.11) [35] with default settings. Additionally, reverse and forward primer sequences were removed with cutadapt (v2.5) [36] with default settings. Sequences were then size filtered (≤300 bp were removed) and dereplicated by vsearch (version 2.14.1) [37]. Denoising was performed with the UNOISE3 module of vsearch and a set minsize of 8 reads. Chimeric sequences were excluded with the UCHIME module of vsearch. This included de novo and reference-based chimera removal against the SILVA SSU 138 NR database for the 16S bacteria and against the SILVA SSU 128 NR database for the 18S eukaryotic [38–40], resulting in the final set of amplicon sequence variants (ASVs). Merged sequences were mapped to ASVs by vsearch with a set identity of 0.97 to construct an abundance table. Taxonomy assignments were performed with BLASTn (version 2.9.0) against the SILVA SSU 138 NR database for 16S, and the SILVA SSU 128 NR database for the 18S with an identity threshold of 100% for the 16S and 90% for the 18S. We used identity and query coverage to mark uncertain blast hits as recommended by the SILVA ribosomal RNA database project with the formula (pident + qcovs)/2 ≤ 93. Amplicon sequence variants were then further analysed and visualised in RStudio using ampvis2 [41] and Phyloseq [42].

*2.6. Statistical Analysis of Data*

Statistical analysis was conducted in R. The two-sample *t*-test was performed using the *t*-test () function with default parameters in order to test whether DNA concentrations differed significantly. A nested data frame approach was then utilised in order to keep the data and test results in the same frame [43]. Data was normalised by rarefaction using MetagenomeSeq [44]. The alpha diversity, using unweighted UniFrac as the distance, was calculated for common metrics (Chao1, ACE, Shannon, Simpson and Fischer) and a Wilcoxon rank-sum test was performed to compare the observed diversity. PCOA using Bray's distances were performed in phyloseq [42], and significance was tested using a permutational ANOVA [42].

*2.7. Sequence Data Deposition*

The 16S and 18S rRNA gene amplicon sequences were submitted to the NCBI Sequence Read Archive4 (SRA) under the NCBI BioProject accession number PRJNA718141.

## 3. Results

The aim of this study was to determine the effect of the DNA extraction method on DNA quality and quantity, as well as how this affects the downstream analysis in microbiome studies through the analysis of samples taken from two ruminally cannulated, non-lactating Holstein Friesian cows. In addition, the extrapolation potential of alternative, less invasive, sampling sites for the cattle rumen were investigated.

*3.1. Evaluation of DNA Extraction Methods*

All sample points and methods yielded high molecular weight DNA suitable for PCR applications, and the use of a bead-beating step greatly increased the concentration of extracted DNA (Figure 1, Table 2). Here, we saw an increase of more than 2-fold for most methods and almost 10-fold higher for the phenol chloroform extraction method when bead-beating was utilised. The integrity of the DNA was highest for the QSB method, where

less DNA shearing and degradation was observed. Statistical analysis was performed using two sample *t*-tests, to confirm that the DNA yield was significantly affected by extraction method ($p < 0.001$). When the complete rumen sample is used as a directly comparative sample, we can see that the best concentrations were collected using the two stool kits (QSB and OS), with bead-beating steps. However, when we take the ratio of absorbance at 260 nm and 280 nm (A260/280), which is an indicator of DNA purity, the purity of the extracted DNA was observed to be higher from the QSB method compared to the OS method (A260/280~1.9 and ~1.7 respectively). The QM method, without the addition of the inhibitor buffer, was unable to produce contaminant free extractions (typically proteins and/or phenolic compounds) that absorb at a slightly higher wavelength than DNA (A260/280 0.8–1.4).

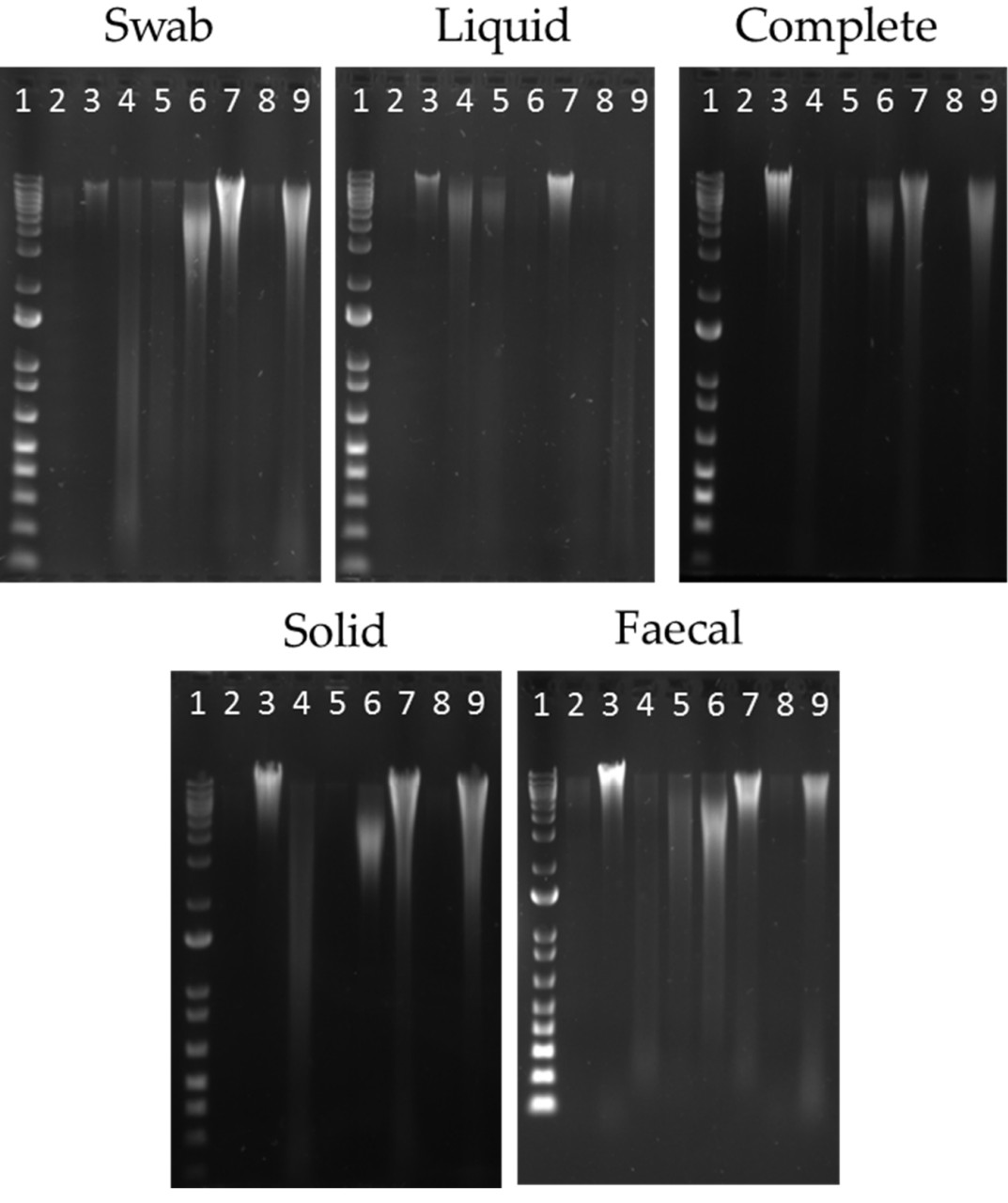

**Figure 1.** Extracted DNA from each sample site; oral swab; ruminal (liquid); ruminal (complete); ruminal (solid); and faecal collection. Molecular marker 1Kb Plus (NEB, UK) shown in Lane 1. Extraction methods shown in Lanes 2–9 as follows; 2-QS; 3-QSB; 4-QM; 5-QMI; 6-OS; 7-RBBC; 8-PC; and 9-PCB.

**Table 2.** Characteristics of DNA isolated with different extraction methods (two sided *t*-test $p < 0.05$) and sample sites (two sided *t*-test $p < 0.001$).

| Method | Swab | Liquid | Complete | Solid | Faecal |
|---|---|---|---|---|---|
| QS | | | | | |
| Concentration (ng/µL) | 1.05 ±1.05 | 22 ±1.5 | 57.75 ±12.05 | 87.75 ±26.1 | 86.6 ±3.4 |
| A260/280 | 2.63 | 2.01 | 1.97 | 1.94 | 1.97 |
| QSB | | | | | |
| Concentration (ng/µL) | 4.15 ±0.65 | 46.65 ±17.25 | 124.1 ±69.6 | 312.05 ±6.05 | 247.65 ±3.25 |
| A260/280 | 1.86 | 2.04 | 1.93 | 1.88 | 1.84 |
| QM | | | | | |
| Concentration (ng/µL) | 4.25 ±1.05 | 84.15 ±3.25 | 107.8 ±67.7 | 248.25 ±39.7 | 142.4 ±13.2 |
| A260/280 | 1.75 | 1.71 | 0.94 | 1.11 | 0.94 |
| QMI | | | | | |
| Concentration (ng/µL) | 4 ±0.6 | 61.05 ±6.65 | 61.25 ±3.95 | 36.35 ±13.25 | 47.5 ±13.5 |
| A260/280 | 1.81 | 1.88 | 1.54 | 1.32 | 1.37 |
| OS | | | | | |
| Concentration (ng/µL) | 3.15 ±2.1 | 102.9 ±16.5 | 228.9 ±19.8 | 299.85 ±44.25 | 152.8 ±4.5 |
| A260/280 | 1.83 | 1.8 | 1.79 | 1.79 | 1.35 |
| RBBC | | | | | |
| Concentration (ng/µL) | 7.44 ±1.6 | 147.3 ±2.6 | 315 ±113.5 | 513.9 ±37.4 | 335.4 ±23.7 |
| A260/280 | 1.83 | 1.8 | 1.57 | 1.58 | 1.35 |
| PC | | | | | |
| Concentration (ng/µL) | 4.72 ±1.48 | 11.75 ±0.45 | 20.85 ±2.75 | 30.85 ±4.55 | 13.5 ±1.1 |
| A260/280 | 1.9 | 1.86 | 1.93 | 1.92 | 1.84 |
| PCB | | | | | |
| Concentration (ng/µL) | 31.3 ±0.7 | 329.5 ±1.7 | 394.35 ±205.3 | 1050.55 ±193 | 834.4 ±51 |
| A260/280 | 2.27 | 1.89 | 1.92 | 1.96 | 1.87 |

A significant difference was also observed through two sided *t*-tests, when comparing extracted DNA concentrations at sample sites ($p < 0.03$). These values were highly variable with oral swab samples, providing much lower DNA concentrations, between 3–10 ng/µL, compared to more than 5-fold higher concentrations in rumen and faecal samples (Table 2). Although well suited for extracting DNA from the oral swab, the QMI method was less successful in extracting DNA from the complete and solid rumen contents and faecal samples. This indicates that the QMI method is better suited to fluid samples (saliva and rumen liquid).

DNA with a A260/A280 ratio between 1.5 and 2 was defined as "acceptable", and those outside this parameter defined as "poor" for further study. Complete rumen samples were used for the amplicon sequencing analysis of the selected extraction methodologies, as previous studies have shown them to be the most indicative of the rumen microbiome [12].

### 3.2. Impact of DNA Extraction Method on Microbial Relative Abundancies

As stated above, the complete rumen sample was used to compare each of the selected extraction methodologies. As the QS, QM and PC methods extracted low yield and poor quality DNA, no further analysis was performed on these samples.

The DNA extraction method was observed to affect the abundance of both the bacterial and eukaryotic taxa at all taxonomic ranks. For the bacterial analysis at the order level, Bacteroida were relatively less abundant by the QMI method and the OS method (7.6 and 25.9%), as compared to the other methods, whereas QSB method had a significantly higher abundance of Bacteroida at (49.7%). In contrast, the taxa Clostridia was significantly more abundant when using the QMI and the OS methods (67.7 and 37.8%), as such, replacing the missing Bacteroida, as the QSB method had a lower abundancy of 19.7%. It is noteworthy that the RBBC method and the PCB method had no significant difference in Bacteroida or Clostridia abundance (40% and 35%, respectively). In fact, the two methods had almost identical abundance ratios, only varying slightly for the Mollicutes abundancies (1 and 0.6%). Actinobacteria abundance varied highly between methods, with the QSB, RBBC and PCB methods observing less than 0.7% (0.11, 0.55 and 0.69%), and the QMI and OS methods were observed at more than double (2.75 and 1.47%). Interestingly, the QSB and OS methods were able to detect a higher abundance of Spirochaetes (4.8 and 8.3%) compared with lower abundancies observed in the other methods (0.2, 3.88 and 3.76%). It was observed that the highest relative abundance of bacteria was unsurprisingly Bacteroidales, closely followed by Clostridiales. These were present in all extraction methods. It was also noted that the two soil kits had a much higher abundance of Fibrobacterales at 14.2 and 24.4%. The most prevalent taxa for the QMI method was Sacchariminodales at 10.1%, compared to >3% for all other methods. It should also be noted that the top 10 bacterial orders made up more than 95% of all reads, with most of the rare ASVs being observed in the remainder (Figures 2a and S1).

For the eukaryotic analysis at the phylum level, Ciliophora were almost absent in the extracted sample using the QMI method (<1%), compared to the other methods. The QSB and the PCB methods yielded a significantly higher abundance of Ciliophora at (71.3 and 76.2%, respectively). Ciliophora were also observed in the OS and RBBC methods (57.3 and 59.9%). In contrast, Streptophyta was significantly more abundant using the QMI method and also observed using the OS and RBBC methods (90.7, 11.1 and 5.5%). Interestingly, the QSB and PCB methods had only a limited abundance of Streptophyta observed (<2%). Species from the Opisthokonts phyla, labelled here as "Fungi" were only observed to be present in using the QSB and the PCB methods (11, 7.7 and 18.8%). Moreover, it should be noted that the QSB method also extracted higher levels of Conosa (7.5%) compared to all other extraction methods (1.6–2%). The observed heatmap for the eukaryotic taxa indicates that the 3 most abundant taxa present in the extracted samples were Litostomatea and Chytridiomycota for most extraction methods, although the QMI method extracted almost 94% Embryophcaea, compared to less than 12% for all other methods. It should also be noted that the top 6 Phylum level taxa made up more than 99% of all reads, with most of the rare ASVs being observed in the remainder/others (Figures 2b and S2).

### 3.3. Impact of DNA Extraction Method on Alpha Diversity and Similarity Analysis

For the bacterial analysis, the alpha diversity metrics indicate that there was no significant difference in the richness and evenness of the populations generated by all methods, apart from the QMI method, which showed a significant lower richness through all metrics ($p < 0.01$) (Figure 3a). When the eukaryotic analysis was evaluated, it indicated that the QSB method generated a significant higher ($p < 0.01$) level of observed richness, compared

to the other methods. The QMI method produced a population that was substantially less diverse than the populations observed from all other methods (Figure 3b).

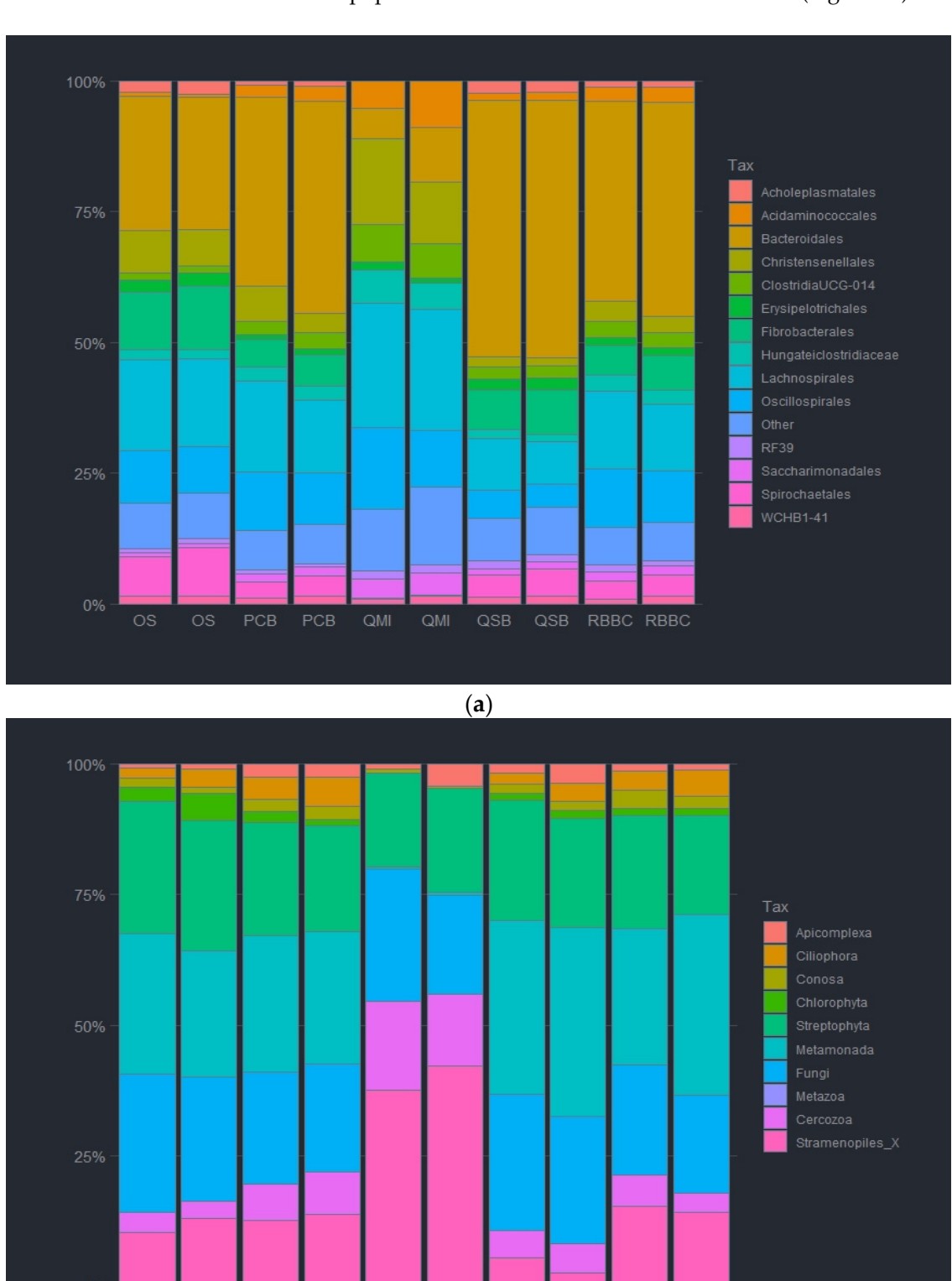

**Figure 2.** Relative abundance of bacterial order (**a**) and eukaryotic phyla (**b**) of ruminal complete sample preparations using different extraction methods: QSB; QMI; OS; RBBC; and PCB.

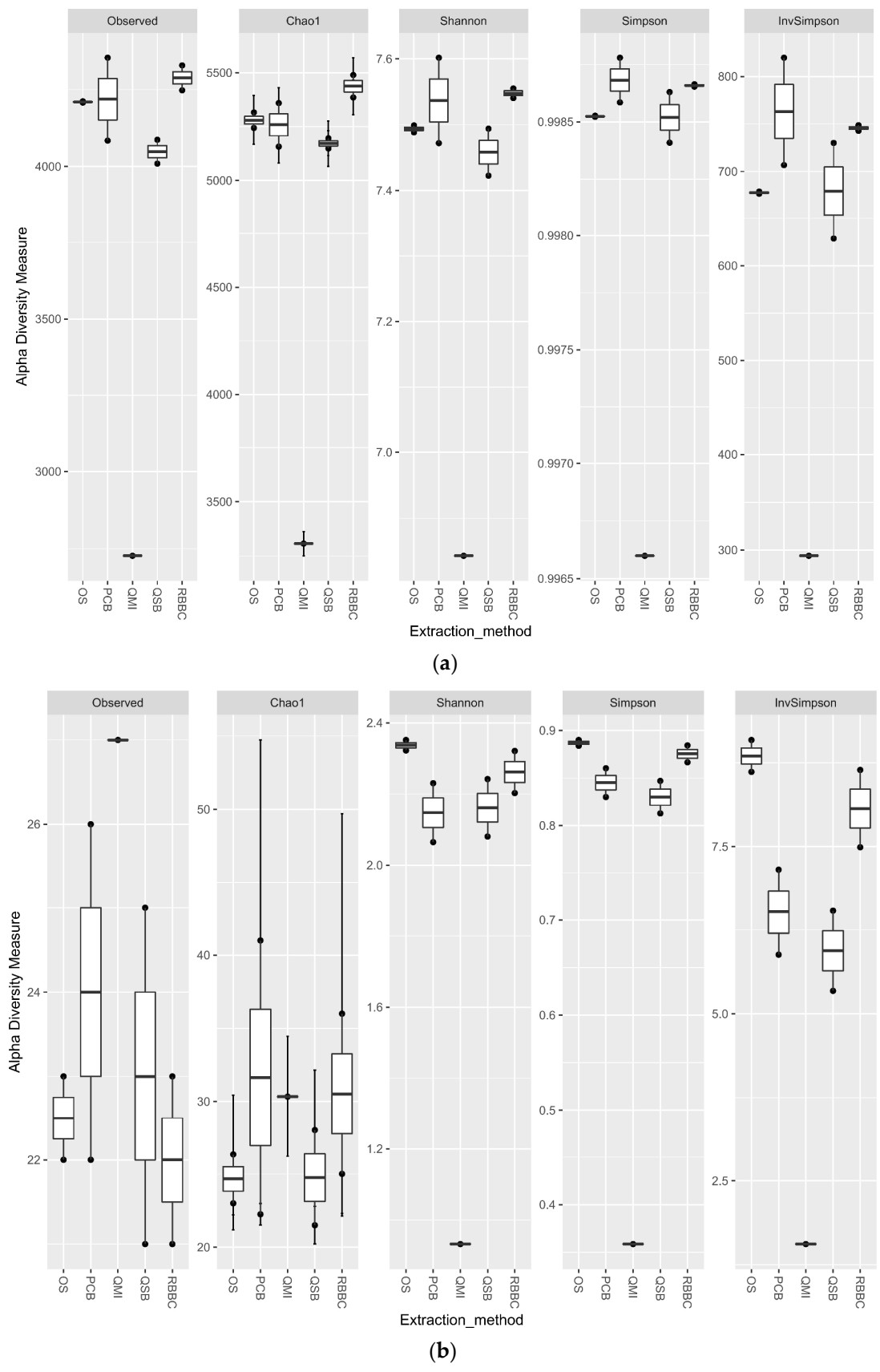

(**a**)

(**b**)

**Figure 3.** Alpha diversity metrics for ruminal sample preparations bacterial (**a**) and eukaryotic (**b**) of ruminal complete sample preparations using different extraction methods: QSB; QMI *; OS; RBBC; and PCB. (Wilcoxon rank sum * *p* < 0.01).

A principal coordinate analysis (PCoA) was used to compare the distance matrix similarities and differences in the microbial community structures of each extraction method. The analysis showed that the QSB, PCB and RBBC methods were closely clustered, indicating that these extraction methods generated very similar community structures, but interestingly using the QMI or OS methods led to significantly variable community structures ($p < 0.01$; Figure 4).

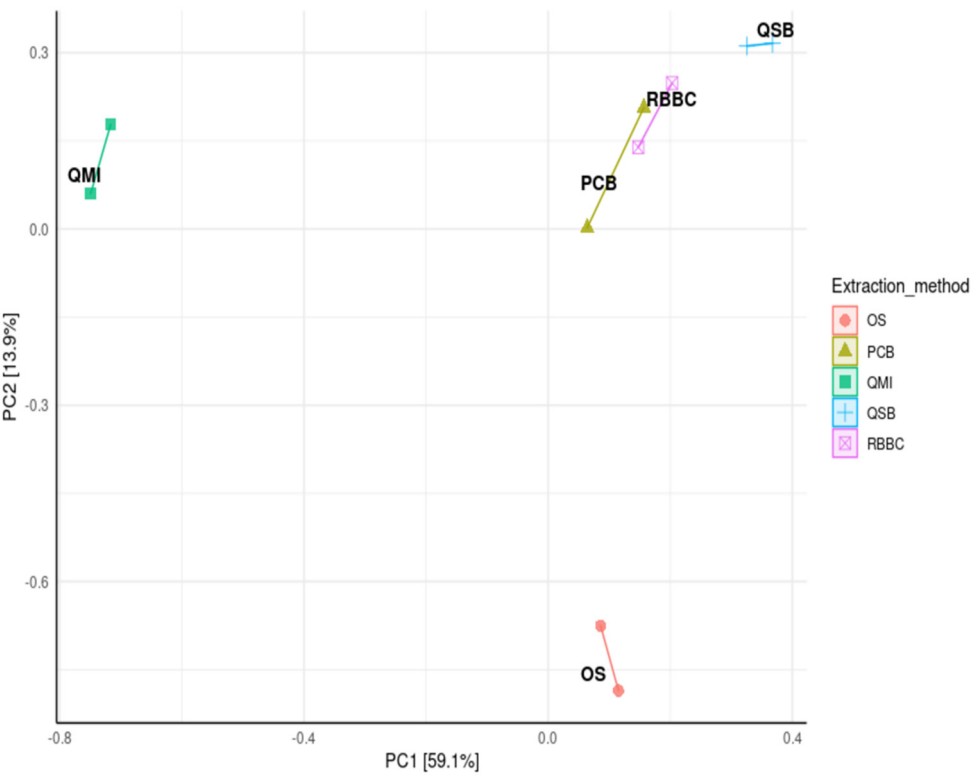

**Figure 4.** Principal coordinate analysis (PCoA), from log transformed data at phylum level of ruminal complete sample preparations using different extraction methods: QSB; QMI *; OS *; RBBC; and PCB, with Bray's distances (permutational ANOVA * $p > 0.01$).

### 3.4. Impact of Sample Site Selection on Microbial Relative Abundancies

All samples were prepared using the QSB method, as this yielded the highest quality DNA, with the least amount of shearing. This standardisation allowed for the removal of biases from different extraction methods, allowing for the data to only reflect changes in the sample site.

The collection site selection affected the abundance of bacterial groups at various taxonomic ranks. At order level (Figure 5a), *Bacteroidia* were less abundant in oral swab samples (2.1%), as compared to the other sample sites (40–52%). *Clostridia* abundance was also reduced, for the oral sample at 0.6%, contrasting the 17.1–20.2% relative abundance observed from the ruminal samples, and 44.9% for the faecal sample site. Over 2% of *Lenistsphaerae* was observed in the liquid and faecal samples, but complete and solid fractions had around 1% relative abundance, while the oral sample had almost none (0.01%). *Tenericutes* abundancy was, interestingly, highest in the liquid rumen fraction (5.5%), with the complete, solid and faecal fractions all having a similar level of abundance (3.3, 2.9, 2.6%), with only 0.02% abundancy being observed in the oral sample. It should also be noted that very low numbers of *Verrucomicrobia* were observed in the oral sample (0.01%), with higher levels of abundance being observed in the rumen fractions, (liquid 0.6%, complete 0.5% and solid 0.6%), which doubled further in the faecal sample to 1.2% (Figure 5a). The heatmap generated from the most prevalent taxa at order level showed that the highest percentage abundance of bacteria was *Bacteroidales*, closely followed by

*Clostridiales* for all sample sites except the oral sample. This was also true for the faecal sample that was made up of over 90% of these taxa. The oral sample was observed to have a higher prevalence of *Pasteurellales* (65.6%) and *Lactobacillales* (26.6%), making up over 90% of the total taxa observed for this sample site. The next top 12 taxa made up the remaining 96% of taxa for the ruminal samples, with the remaining taxa (summarised as "Other") being combined to indicate rare ASVs (Figures 5a and S3).

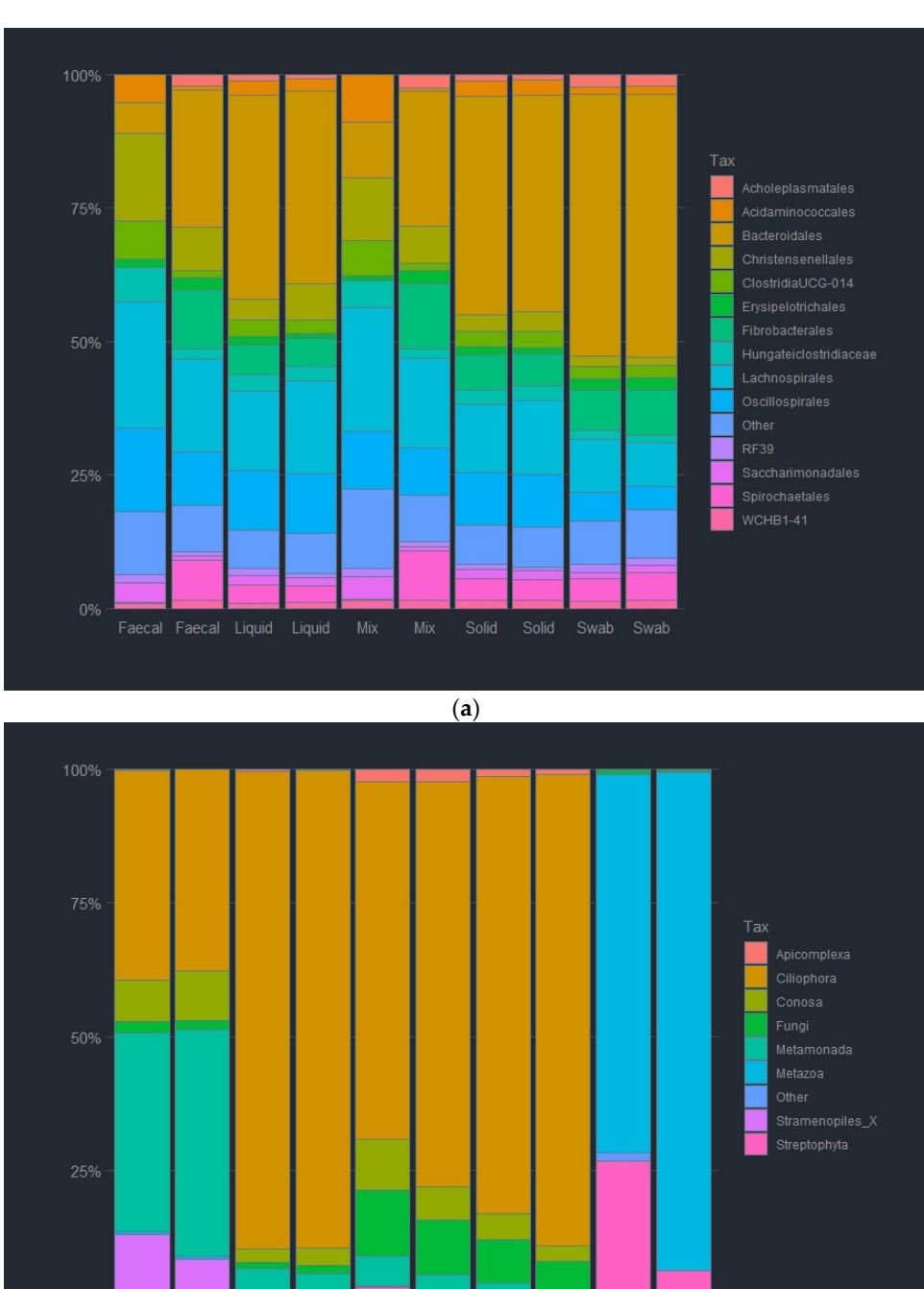

**Figure 5.** Relative abundance of bacterial order (**a**) and eukaryotic phyla (**b**) of sample preparations from different sample sites; oral swab; rumen fraction (liquid); rumen fraction (complete); rumen fraction (solid); and faecal collection.

Similar to the bacterial community composition, the collection sites affected the composition of eukaryotic groups at various taxonomic ranks. At the phylum level (Figure 5b), *Metazoa* were significantly more abundant in the oral swab sample, making up 84% of the observed amplicons, compared to the other sampling sites (>1%). *Ciliophora* abundance, however, was notably reduced for the oral sample at <1%, contrasting to 71–89% abundancy observed from the rumen sample site. The faecal sample also had higher numbers of *Ciliophora* at 37%. It was also observed that higher numbers of *Metamonada* were present, with the faecal sample making up 39% of the total observed counts, as opposed to the rumen samples that had between 1.8 and 6.3%. The faecal fractional also has a higher abundancy of *Stramenopiles* than any other sampling sites at 10.2%. The rumen fractions all had a similar eukaryotic pattern, with only the concentrations present varying slightly between sample, all observed the presence of *Ciliophora*, *Metamonada*, *Menoza* and *Streptophyta*, albeit in slightly varying concentrations. The liquid fraction had the highest levels of *Ciliophora* at 89%, compared to the complete and solid fractions (previously stated). The complete fraction showed higher concentrations of both *Archamoebea* and *Chytridiomycota* at 7.5 and 11.1%, compared to <4% *Archamoebea* abundancy, and <7% *Chytridiomycota* abundancy for both the liquid and solid fractions. The heatmap indicated that the most prevalent taxa at order level were *Litostomatea* for the ruminal samples (73.8–91.9%), with the faecal sample only having 37.7% present. The faecal sample also had a high percentage of *Parabasalia* taxa (39.4%) compared to all other samples (<6%). The oral sample had a large *Cramiata* component (83.2%), with another 14.4% being made up of *Embryophyceae*. The remaining 2.4% were then made up of the remaining ("Other") taxa. Over 10% of the faecal sample was also made up of *Opilinata* and *Archamoeba* (8.5%). *Chytridiomyota* was observed in the complete and solid fractions (11.9 and 6.8%, respectively), with *Archamoeba* also present (6.9 and 3.9%) (Figures 5b and S4).

### 3.5. Impact of Sample Site Selection on Alpha Diversity and Similarity Analysis

Comparing the alpha diversity metrics for the sample sites, indicated that all of the ruminal sites generated highly diverse samples. The faecal sample was less diverse than the ruminal samples ($p < 0.1$), but had a higher observed richness and evenness than the swab site. The swab sample site was significantly less diverse than all other sample sites ($p < 0.02$) (Figure 6a). The trend for the richness levels continued in the eukaryotic samples, with all of the ruminal sites generated highly diverse samples. The faecal sample was equally as rich for the eukaryotic sequencing as the ruminal samples, but the richness was significantly lower for the oral swab site ($p < 0.01$) (Figure 6b).

A principal coordinate analysis (PCoA) was used to compare the distance matrix similarities and differences in the microbial community structures of each sample site. The analysis showed that the ruminal samples were closely clustered, indicating that these sites contained very similar community structures. It was also observed that the faecal and oral samples had very distinct but separate community structures with significant level of variability ($p < 0.001$ respectively; Figure 7).

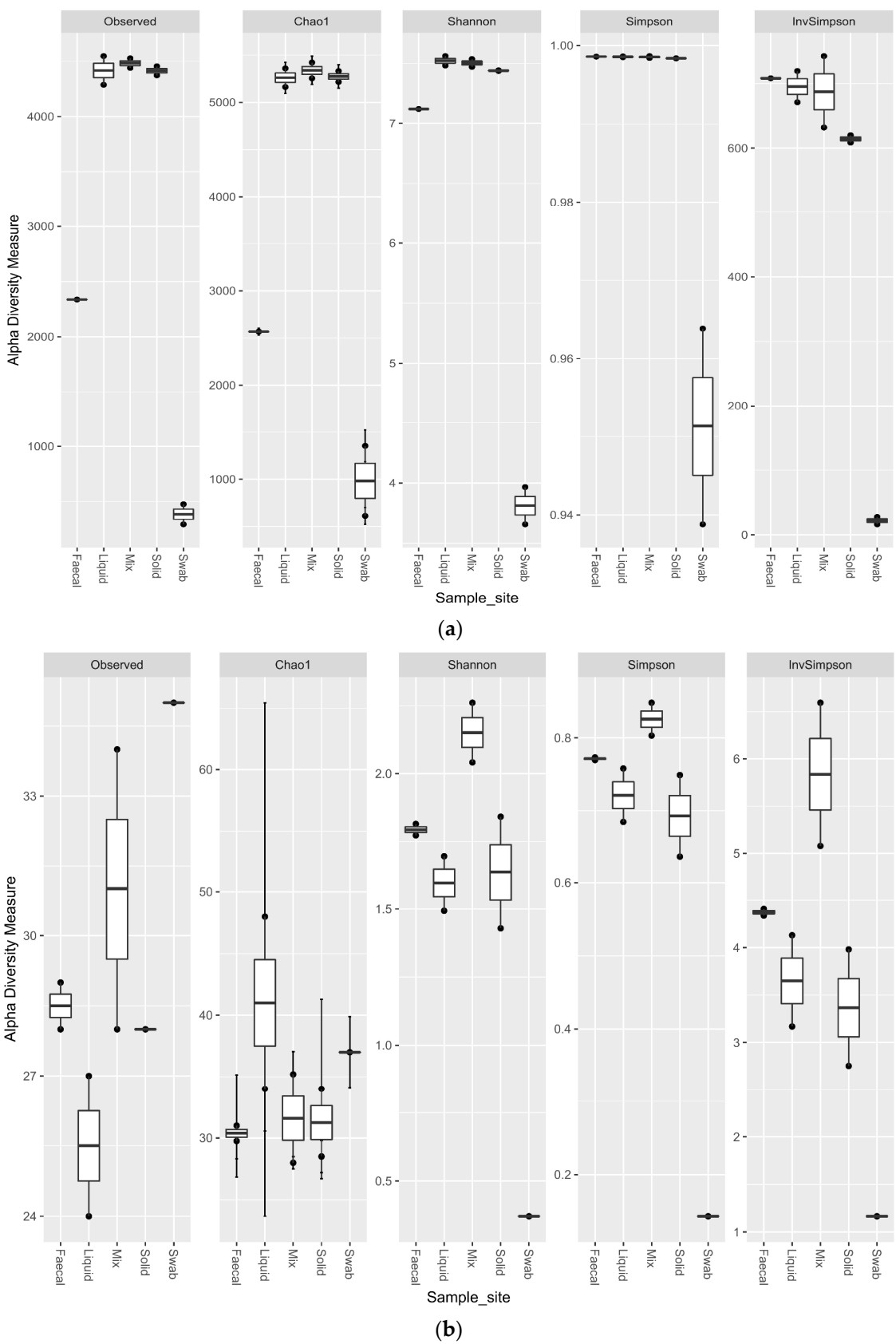

**Figure 6.** Alpha diversity metrics for ruminal sample preparations bacterial (**a**) and eukaryotic (**b**) of ruminal sample preparations from different sample sites; oral swab *; rumen fraction (liquid); rumen fraction (complete); rumen fraction (solid); and faecal collection *. (Wilcoxon rank-sum test * $p < 0.01$).

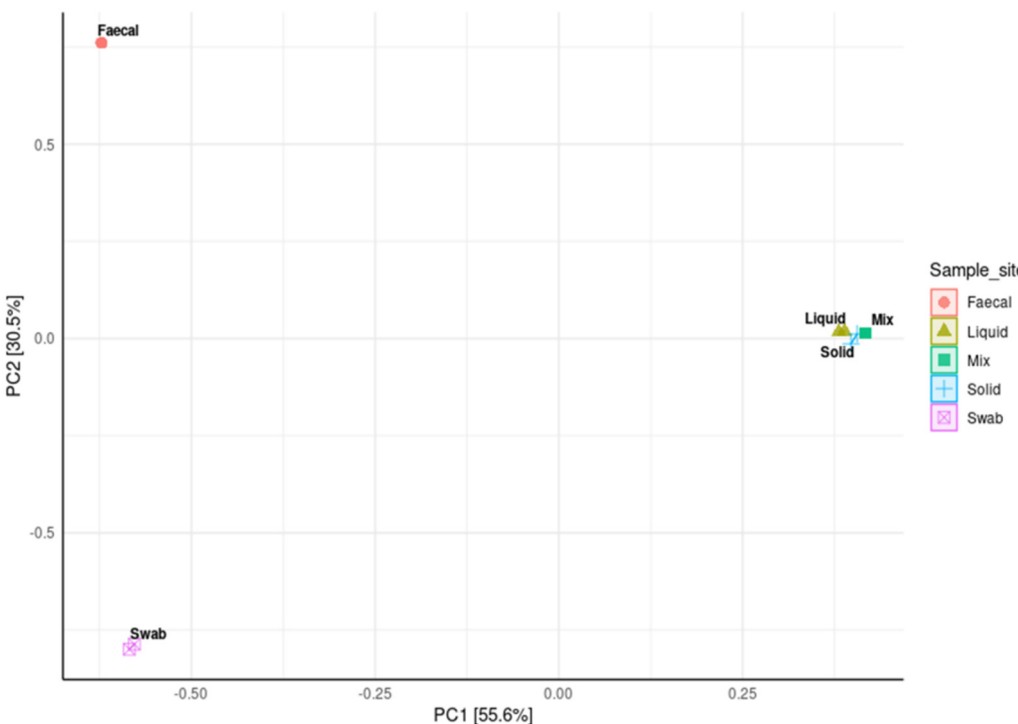

**Figure 7.** Principal coordinate analysis (PCoA) from log transformed data at phylum level of ruminal sample preparations from different sample sites; oral swab *; rumen fraction (liquid); rumen fraction (complete); rumen fraction (solid); faecal collection *, with Bray's distances. (Permutational ANOVA * *p* < 0.001).

## 4. Discussion

There is a pressing need to carry out large-scale ruminal microbiome analysis, to generate a microbial phenotype for use as a trait in future animal selection for enhanced rumen function, including reduced methanogenesis [45,46] and feed conversion efficiency [45–47]. Both methanogenesis and feed conversion efficiency have major implications on the environmental impact of ruminant production [45]. The microbiome phenotype can be defined as a simple ratio [47], or be far more complex based on microbial gene abundance [48].

This investigation examines the possibility of using alternative sampling sites in order to negate the necessity of invasive ruminal sampling. It has been shown that there is a high level of difference between oral, faecal and ruminal sample microbial communities, but that the faecal sample shows some general similarities with the rumen fractions. Many studies examined the microbiota of the rumen and some characterised the faecal community. Relatively few, however, have compared the microbiome obtained from the two different sampling sites directly on the same cohort of animals. Although not directly comparable with 16S rRNA amplicon sequencing, several studies have used restriction fragment analysis to obtain a fingerprint of each type of sample, showing marked differences in the gross structure of the two communities [49,50]. Although the ruminal microbiota contains mainly bacteria, archaea, anaerobic fungi and ciliate protozoa are also present. The bacterial communities observed here were similar to those previously analysed, being dominated by *Firmicutes* and *Bacteroidetes* [13]. The ruminant faecal microbiota has been previously examined [49,50], with results similar to those presented here, in which *Bacteroidetes* are less abundant in faecal samples compared to the ruminal samples. It was hoped that, by remaining at the order and phyla levels for the bacterial and eukaryotic analysis, we would be able to observe some level of corresponding microbial presence that can be attributed back from the faecal samples to the ruminal samples. Whilst they held a general similarity the microbiomes varied between sample sites, there was obvious differences in the relative abundancies. This is most likely due to the change in roles from fore-gut to hind-gut. It

has been previously suggested that whilst the rumen is responsible for the decomposition of plant cell materials [51], this is an essential step in the absorption of nutrients in cattle, in which higher cellulolytic activity can lead to higher representations of *Bacteroidetes* and *Spirochaetes*. This differs from the hind-gut, in which the introduction of fluid and bile aid the intestines in breaking fat, protein and starch whereas higher proteolytic activity present in the hind gut can lead to the dominance of *Firmicutes* and *Proteobacteria* [49,52]. The ratio of *Bacteroidetes* to *Firmicutes* has been observed to alter due to changes in feed [53]. Faecal sampling can be a non-invasive way of attempting to link changes in bacterial abundancies to further functional traits. Furthermore, from a breeding perspective, the observation of core species through faecal sampling can enable hologenomic selection approaches to be utilised in cattle [23]. As such, the use of faecal samples to generate microbial similarity matrices can be more effective than those from oral samples due to the need for a high level of reproducibility and repeatability. For oral sampling, this can pose some difficulties due to the need to capture samples close to rumination for accurate results [16]. This is not an issue that faecal sampling faces. This is, however, beyond the scope of this study and would have to be proven in a large number of animals.

Most notably though, the proteobacterial species were highly abundant in the oral samples making up over 65% of the overall oral microbiome observed here. They represent the oral bacteria most commonly found as part of a distinct gingival microbiota [16]. This would have been compounded due to the presence of bovine saliva, which would inevitably dilute the residual bolus sample, thus preventing a true estimation of rumen microbial groups present. This is most obvious when comparing the oral sampling against any of the rumen samples collected, in which the communities are clearly highly variable. The suitability of buccal swabs or regurgitated digesta as an alternative to rumen sampling will depend on the microbial communities being investigated and the line of scientific enquiry. However, the observations here would indicate that the general oral microbiome was not indicative of the rumen microbiome as such. This result contradicts previous works, which suggest a distinct similarity between the oral and ruminal microbiomes can be observed [13,16]. However, a large amount of data manipulation, through bioinformatics depletion, was undertaken in these studies to remove the oral bacteria not representative of the rumen. This produces a number of issues, most notably reducing the sequencing depth of the study, but also relying heavily on the researcher being confident that only non-rumen bacteria are removed. The more steps involved in the processing of microbiome samples the more at risk of introducing compound errors to these studies. As this article has shown, each step, from sample collection to processing methods and beyond, can introduce errors and adaptations that can be compounded throughout the data processing steps to the observable results. As such, the data presented in this study indicates that as a direct proxy, the oral microbiome has limited value as a biomarker of the rumen microbial community without a high level of post sequencing manipulation. Another factor potentially affecting the output is sample collection. Unless the sample is procured directly after rumination, when ruminal digesta is present, this will have a large effect on the observed results. Whilst reducing bias during sampling, other limitations can occur. The ruminal bacteria regurgitated in the bolus only have the ability to last a limited amount of time within the oral cavity due to the oxygen rich environment, which is detrimental to strict anaerobes, such as the methanogens and protozoa, present in the bolus [54]. As such, if rumination has not recently occurred, then oral samples will be limited in their ability to indicate the microbiome of ruminal fractions. This can also be compounded a dilution effect of the bovine saliva, although this may be less relevant directly after rumination. It is very clear that the timing of sampling from the oral microbiome can give highly variable results, with special relevance placed on the presence or absence of the bolus.

The choice of primers can also affect the end results. Due to technical constraints we chose to use a generalised 18S primer pairing that covers V4 rRNA loop with the aim of capturing a broad snapshot of the rumen. However, it has been shown that perhaps a more focused choice of specifically archaea, ciliate or fungal primers would aid in the

elucidation of the rumen microbiome [55–57]. Here, we aimed to show what similarities can be observed between the sample sites as well as highlighting the differences caused by various extraction methods. As such, the generalised nature of the 18S primers used in this study allowed for the observation of a wide spectrum of over 1000 potential species to be noted.

Typically, the influence of environment, specifically diet, on the composition and function of the ruminal microbiome [16,50,58], and direct or indirect associations with animal performance traits [9] have been investigated. The insights given here promote the use of less invasive sampling methods, which can be carefully utilised to help compare and evaluate their impact on functional traits. Specific environments, such as the mouth, seem to harbour a set of specific microorganisms that can influence co-occurrence results when compared to rumen composition. Most ruminal microbiota analysis has been descriptive rather than predictive, although major studies are under way to change the situation regarding predictive approaches for methane emissions and feed efficiency [9,47].

In this study, we also investigated the impact and possible biases caused by different DNA extraction methods. Although it was not possible to determine which method extracted DNA most representative of the rumen microbial community, the data shown here does indicate that as expected, methods without a mechanical lysis step extract less DNA, and therefore limiting microbial diversity observed. This would be in agreement with previous studies that have indicated that mechanical lysis is necessary to open up Gram-positive bacteria, such as *Firmicutes*, which are difficult to lyse [59]. Henderson et al. (2013) have also indicated, however, that increasing the length of the bead-beating step can also lead to biases in the bacterial microbiome extracted [12]. They showed that through the utilisation of the bead-beating step, the abundance of *Firmicutes* increased, but the abundance of *Bacteroidetes* reduced. This can be somewhat explained by the compositional nature if the data collected. However, both *Firmicutes* and *Bacteroidetes* have been linked in studies to traits, such as average daily gain, and have the ability to utilise differing carbohydrate sources [60], making them of high importance in future microbiome studies. As eukaryotes and archaea are important in ruminal studies, methods that contain host DNA removal steps should be avoided. As was observed in this study, the QMI method is one such method, which contains an initial treatment with a specific buffer and benzonase. This led to the suppression of our eukaryotic community, where less alpha diversity was observed in the samples. In a ruminal study this will greatly affect the results produced, skewing the data collected. If variations occur that are linked to the extraction method rather than the study itself, then this will impact the conclusions. This trade off will need to be addressed when planning future studies, between the need for mechanical lysis and the possibility of damage to the DNA itself from both shear forces and from heat generated by the process. Obtaining sufficient DNA that is representative of the rumen microbial community without damaging the DNA in the process of extraction is a delicate balance. It should also be stated that whilst *Cilates* have been observed to make up around 50% of the ruminal fraction, ranges of more than 50–70% were observed in our samples [61]. This can be explained by only comparing the observed eukaryotic amplicons against each other, and not as part of the combined rumen microbiome as a whole. With *Cilates* predominantly consuming bacteria as their main protein source, any variations observed can indicate potential defaunation events; however, this would have to be linked to the microbiome as a whole [61,62]. However, as the nature of the data is compositional, it is important to state that variations in one taxa will cause changes to all the remaining taxa, as overall percentages are affected [63].

Multiple factors, such as effectiveness, cost and user safety, can all play a role in the selection of DNA extraction methods for microbiome studies. Although relatively cost effective, and able to generate high quality DNA, the PCB method should be used cautiously, as the components are dangerous to handle, and should only be performed by trained researchers in the methodology. The kit-based methods (OS and QSB) are much safer to use but have higher cost implications attached. It is important to note that the RBBC

method can be viewed as a variant of these methods, but with the bonus of having buffers that are easy to produce. A multi-faceted approach needs to be taken by the researcher when choosing a method to work with, taking into account the funding, the researchers abilities and the experimental environment.

It is important to note, that this study was performed using a relatively small sample number (n = 2), and as such can only give an indication of statistical relevance. The results presented here therefore should only give more weight to the need to have a fully standardised method of ruminal microbiome studies in order for accurate data to be obtained.

### 5. Conclusions

Each method used to extract community DNA will lead to a difference in observed relative abundances of bacterial and eukaryotic community compositions. This can lead to misleading interpretations when attempting to compare separate studies, which utilised different methods. From the methods utilised in this study, we would recommend the use of the QSB method as an efficient and reproducible method for DNA extractions in rumen studies. There is also a definite need for alternative, less invasive, sampling methods of ruminants to help link the microbial community to functional traits. As such, faecal sampling could be used for this type of analysis, but this must be approached cautiously, as there are limits to the depth of community analysis possible without direct rumen sampling due to a reduction in the observed diversity.

**Supplementary Materials:** The following supporting information can be downloaded at: https://www.mdpi.com/article/10.3390/ruminants2010007/s1, Figure S1. Heatmap of relative abundance of bacterial order (a) and eukaryotic phyla (b) of ruminal complete sample preparations using different extraction methods; QSB; QMI; OS; RBBC; PCB. Figure S2. Heatmap of relative abundance of eukaryotic phyla of ruminal complete sample preparations using different extraction methods; QSB; QMI; OS; RBBC; PCB. Figure S3. Heatmap of relative abundance of bacterial order sample preparations from different sample sites; Oral swab; Rumen fraction (liquid); Rumen fraction (complete); Rumen fraction (Solid); Faecal collection. Figure S4. Heatmap of relative abundance of eukaryotic phyla of sample preparations from different sample sites; Oral swab; Rumen fraction (liquid); Rumen fraction (complete); Rumen fraction (Solid); Faecal collection.

**Author Contributions:** Conceptualisation, A.C.M., M.H., J.H. and J.T.; methodology, A.C.M., M.H.; bioinformatic analysis, D.S.; formal analysis, A.C.M.; investigation, A.C.M.; resources, J.T.; data curation, D.S.; writing—original draft preparation, A.C.M.; writing—review and editing, A.C.M., D.S. and J.T.; visualisation, A.C.M. and D.S.; supervision, J.T.; project administration, J.T.; funding acquisition, J.T. All authors have read and agreed to the published version of the manuscript.

**Funding:** This research received no external funding.

**Institutional Review Board Statement:** The study was carried out in accordance with the EU Directive 2010/63/EU for animals used for scientific purposes and the Council for Animal Welfare at the University of Göttingen has approved the study (18A269).

**Informed Consent Statement:** Not applicable.

**Data Availability Statement:** The datasets generated for this study are available on NCBI BioProject accession number: PRJNA718141.

**Acknowledgments:** The authors would like to acknowledge the G2L: Next-Generation Sequencing (NGS) laboratory in Göttingen who performed the rRNA sequencing.

**Conflicts of Interest:** The authors declare no conflict of interest.

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
