# Peer review of "Bovine Rumen Microbiome: Impact of DNA Extraction Methods and Comparison of Non-Invasive Sampling Sites"

_ruminants, doi:10.3390/ruminants2010007_

Round 1

Reviewer 1 Report

Overall a relevant topic covering some important aspects of rumen function. However, careful efforts are needed to revise different sections of this paper especially the experimental design and replications concerning sampling frequency and procedures of data analysis that were employed in this study. please check the accuracy, completeness and consistency of both cited and listed references while revising this paper.

Author Response

Dear reviewer,

Thank you for taking the time to review the article and for your comments

We have made a number of changes to each section which can be found in the updated manuscript, as well as reviewing and revising the citations referenced here. We hope that these adaptations are adequate for your approval.

Reviewer 2 Report

The aim of this study was to investigated the impact of different DNA extraction methods on cattle ruminal microbial community composition. For this purpose, five DNA extraction methods were used, showing a high level of variability in relation to the microbial communities observed.

In addition, the authors also investigated possible alternative non-invasive sampling sites. Therefore, oral swabs and fecal samples were taken and their microbiomes were compared to the rumen one.

First of all, the title does not address the topic of the manuscript adequately. Cattle microbiome is a generic word that can refer to gut microbiome as well as milk microbiome or vaginal microbiome. In my opinion, therefore, the title must be changed to “Cattle rumen microbiome: etc etc” or “Bovine rumen microbiome: etc etc”.  The same in the Abstract.

Furthermore, the Introduction does not include all relevant references. For example, at line 63, references of other papers such as Wallace et al. GSE 2017, Fatehi et al. JDS 2015 and Fliegerova et al. Anaerobe. 2014, are missing. Tapio et al. PlosONe 2016 is present in the References (17) but is missing here.

Finally, I have some doubts that the research design was entirely appropriate in considering faecal samples as an alternative to ruminal ones. Usually, as supported by a lot papers, fecal samples were used as a non-invasive alternative to gut samples. But the rumen has different functions from the other intestinal tracts and consequently a different microbiome, as recently described in many papers, such as in Animal (2018), 12(2), s220-s232, in PLOS ONE (2020), 15(4), e0231533, in GigaScience (2020),9,6: giaa057 and in Polish journal of microbiology (2021), 70,2: 175-187.  Therefore, in my opinion, the authors should better detail their choice.

Author Response

Dear reviewer,

Thank you for taking the time to review the article and for your comments

Firstly we have altered the title to clearly reflect the topic, the title now reads “Bovine rumen microbiome: Impact of DNA extraction methods and comparison of non-invasive sampling sites”.

The relevant references have now been added to the introduction.

Thank you for your comment. We agree that it might be misleading to talk about faecal samples as a true alternative to rumen samples. This was not our intention. From a breeding perspective, one needs a non-invasive sampling method that can be conducted in many animals. In hologenomic selection approaches, it is necessary to derive a microbial similarity matrix that is based on measures with high repeatability. As the outcome of oral sampling is dependent on the exact time of rumination, the measures might not be very repeatable. As these approaches are based on explained variances, the faecal microbiota on the other hand might be as suitable as the ones found in the oral cavity, although they are of course not a true proxy for the ruminal microbiota. This is, however, beyond the scope of this study and has to be proven in a large number of animals. We have explained this in more detail in the manuscript now in order to make the selection of sampling sites clear. Please refer to lines #88-93 in the introduction and discussion line # 550-571.

We hope that these adaptations to the manuscript are adequate for your approval.

Reviewer 3 Report

Dear authors,

The objective of this study was to compare the different DNA extraction methods on microbial composition (relative abundances) and also to choose the proxy for rumen samples. There have been several studies published on this topic. In this study authors have chosen several extraction methods to extract the DNA from different types of samples. This is an interesting study. But, I have some major concerns on the methods and the interpretation of the results.

  • You have chosen oral and faecal samples as the proxy for the rumen sample. From my point of view, oral samples can be used as the proxy for rumen samples, but faecal samples can be used as the proxy for hind-gut, not for rumen.
  • You have mentioned that you collected buccal fluid. Is this sample contaminated with saliva? I do not see more details about the collection method. Did you swab the buccal? How long did you keep the swabs inside the mouth? As several studies showed that the oral samples can be used as the proxy for rumen samples (Kittelmann et al., 2015 and Tapio et al., 2016), you need to mention the collection and processing methods in detail. According to your method, you centrifuged the swabs to get the liquid out? Did you think that the bacteria attached to the swabs can be removed by this?
  • Kittleman et al., (2016), removed the oral bacteria from the buccal swabs bacterial community during sequence analysis and find out that the oral samples can be used as the proxy for the rumen samples? May be you can check this?
  • I am not convinced the primers chosen for eukaryotic 18S rRNA gene amplification. Did you check the suitability of this primer for rumen samples? Many of the published studies in rumen sample used 18S rRNA gene for protozoal composition and ITS region for fungal composition. Is 18S rRNA gene is appropriate for rumen fungal composition. To compare different extraction methods the choice of the primer is ok, but to check different sites, I think the primer choice is important.
  • The details of the sequence analysis is too brief. I am not sure why did you use same processing parameters for both bacteria and eukaryote? Especially identify threshold of 90% for both bacteria and eukaryote? Is that too low for bacteria?
  • Bacterial composition was given at order level? For good comparison, the bacterial composition can analyzed at genera level. I am not sure why authors stopped at order level for bacteria and phylum level for eukaryote.
  • The quality of the figures are not good and unreadable. It is better to give the relative abundances of different taxa in tables rather than in figures. And please incorporate statistical significances in this tables.
  • I do not understand the statistical analysis? Why did you use 2-sample t-test? I do not think this is an appropriate statistical test to compare different extraction methods and different sampling sites? Also, the statistical significances were only mentioned in the main text of the manuscript, not in any figures or tables. These statistical significances should appear in the tables and figures.
  • Authors also have to be careful of the spelling of the taxa. For example class Bacteroida, spelled wrongly in many places. Also authors described the bacterial relative abundances at class level in 305-326, I do not see these values and statistical significances in any of the tables or figures.
  • Why did you do the PCoA plot at phylum level. It is better to do this at ASV level.

  • Please describe the fecal sample collection method properly. From where did you directly grab the sample and how much did you collect?

  • Why did not you do qPCR to check the absolute abundances of different microbial taxa?

  • Is fungi is eukaryotic phyla?

  • In the discussion part, I did not see any discussion about fecal microbial community.

  • Surprisingly, the microbial composition of 2 cows is very similar in all sites. This may be because, you analyzed them in order and phylum level. Definitely there will be individual variations,  which can be observed at genera level.

Specific comments

Line 118: three sample types: fistula collection is the method, not the sample type. You can mention as composite rumen samples, buccal fluid and faces.

Lines 307- 309: spelling Bacterioda

Line 305: you only studied relative abundances

Line 305-326: you are talking here the bacteria in class level, but I do not see these information anywhere in the figure. Furthermore the figure does not give the information about the relative abundances and statistical differences. The figure is in low quality. Give these information in tables, preferably for bacteria at genera level?

Line 351: alpha diversity: you studied both richness and evenness.

Author Response

Dear reviewer

Thank you for taking the time to review the article, for your comments and for giving us the opportunity to resubmit our manuscript.

We have attempted to respond to each of your comments individually and have given a numerical designation to each. Please find below the detailed response, and we hope that these adaptations are adequate for your approval.

  1. We agree that it might be misleading to talk about faecal samples as a true alternative to rumen samples. This was not our intention. From a breeding perspective, one needs a non-invasive sampling method that can be conducted in many animals. In hologenomic selection approaches, it is necessary to derive a microbial similarity matrix that is based on measures with high repeatability. As the outcome of oral sampling is depending on the exact time of rumination, the measures might not be very repeatable. As these approaches are based on explained variances, the faecal microbiota on the other hand might be as suitable as the ones found in the oral cavity, although they are of course not a true proxy for the ruminal microbiota. This is, however, beyond the scope of this study and has to be proven in a large number of animals. We have explained this in more detail in the manuscript now in order to make the selection of sampling sites clear. Please refer to lines #88-93 in the introduction and discussion line # 550-571.

  1. The oral swab methodology has been updated to cover the points that you have made:

“The buccal fluid collection was performed as follows: The animals were incentivised to bring the bolus forward with feed being placed just out of reach. The sterile salivette® collection cotton swabs (Sarstedt, Germany) were then placed in the mouths of the cows using sterile forceps and rubbed for 1 minute within the mouth (on the walls).  1mL of PBS solution was added to the swab, which was vigourously agitated at 50 °C in an orbital shaker (Sartorius) for 1 h. The swabs were then centrifuged at 10,000 x g to remove all collected sample from the swab and the bacteria was eluted and collected. This was then used in the further extraction method. This method allowed for a reasonable collection of saliva and oral microorganisms similar to previously published methods [16,26]”

  1. Although very interesting and pertinent to this study, the work of Kittleman et al requires a large amount of data manipulation, through bioinformatics depletion, removing the oral bacteria not representative of the rumen. This produces a number of issues, most notably reducing the sequencing depth of the study, but also relying heavily on the researcher being confident that only non-rumen bacteria are removed. The more steps involved in the processing of microbiome samples the more at risk we are of introducing compound errors to these studies. We have therefore chosen not to take this route of action. As such, we believe that the data presented in this study indicates that as a direct proxy, the oral microbiome has limited value as a biomarker of the rumen microbial community without a high level of post sequencing manipulation. Also see lines #582-590 in the revised manuscript.

  1. The choice of primers can also affect the end results. Due to technical constraints we chose to use a generalised 18S primer pairing that covers V4 rRNA loop with the aim of capturing a broad snapshot of the rumen. However, it has been shown that perhaps a more focused choice of specifically archaea, ciliate or fungal primers would aid in the elucidation of the rumen microbiome [49–51]. Here we aimed to show what similarities could be observed between the sample sites as well as highlighting the dif-ferences caused by various extraction methods. As such the generalised nature of the 18S primers used in this study allowed for the observation of a wide spectrum of over 1000 potential species to be noted. Line #607-615.

  1. Apologies, the bacterial analysis was indeed performed at a threshold of 100% please see revised paragraph as have made additions to the methods section from both bacterial and eukaryotic processing:

Demultiplexing and clipping of adapter sequences from the raw amplicon se-quences were performed with the CASAVA software (Illumina). The program fastp (v0.20.0) [31] was used for quality filtering with a minimum phred score of 20, a minimum length of 50 base pairs, a sliding window size of four bases, read correction by overlap and adapter removal of the Illumina Nextera primers. Paired-end reads were merged with the paired-end read merger (PEAR v.0.9.11) [32] with default settings. Additionally, reverse and forward primer sequences were removed with cutadapt (v2.5) [33] with default settings. Sequences were then size filtered (≤300 bp were re-moved) and dereplicated by vsearch (version 2.14.1) [34]. Denoising was performed with the UNOISE3 module of vsearch and a set minsize of 8 reads. Chimeric sequences were excluded with the UCHIME module of vsearch. This included de novo and reference-based chimera removal against the SILVA SSU 138 NR database for the 16S bacteria and against the SILVA SSU 128 NR database for the 18S eukaryotic [35–37] re-sulting in the final set of amplicon sequence variants (ASVs). Merged sequences were mapped to ASVs by vsearch with a set identity of 0.97 to construct an abundance table. Taxonomy assignments were performed with BLASTn (version 2.9.0) against the SILVA SSU 138 NR database for 16S and the SILVA SSU 128 NR database for the 18S with an identity threshold of 100 % for the 16S and 90% for the 18S. We used identity and query coverage to mark uncertain blast hits as recommended by the SILVA ribosomal RNA database project with the formula (pident + qcovs) / 2 ≤ 93. Amplicon sequence variants were then further analysed and visualized in RStudio using ampvis2 [38] and Phyloseq [39].

  1. Whilst the depth of the bacterial comparison could have been increased to genera level, due to the low sample size (n=2) we aimed to focus the resolution of the study on the observable differences from sample site and extraction method.

  1. The uploaded figures meet the requirements of the journal (1000 dpi), please find in supplementary information the heatmaps S1-S4.

  1. After review, and due to the limited sample number and the compositional nature of the abundancy data, true significance cannot be drawn here, these values have therefore been removed from the article.

  1. Thank you for pointing out this error, we have been through the article and have now corrected these errors.

  1. As stated above, due to the low sample number the decision was made to focus on the level where the observable differences could be clarified.

  1. Please see lines #140-145 “The faecal samples were collected by stimulating rectal activity in order to generate fresh material. Faecal samples were taken approximately 60cm deep inside the rectum using sterile gloves. Approximately 500g of faecal matter was collected, from which a smaller sample (50g) was taken for this study. These samples were then taken for further evaluation and placed in a sterile container. To avoid any process time bias all samples were stored at -80 °C directly after collection. ”.

  1. Although the use of QPCR would give a more robust data point, the relative abundancies meet the general aim of this study, attempting to elucidate the impact of the extraction method and sample site. Here I again point out that this study is attempting to show where possible similarities can diverge from one another due to these parameters.

  1. The fungi refered to in this graph are opisthokonts who are linked to both metazoans and fungi, they were observed in our samples and marked as such. A clarifying sentence has been added at line #361

  1. Please find the following paragraph in the discussion #539-571

” Many studies examined the microbiota of the rumen and some characterised the faecal community. Relatively few, however, have compared the microbiome obtained from the two different sampling sites directly on the same cohort of animals. Although not directly comparable with 16S rRNA amplicon sequencing, several studies have used restriction fragment analysis to obtain a fingerprint of each type of sample, showing marked differences in the gross structure of the two communities [49,50]. Although the ruminal microbiota contains mainly bacteria, archaea, anaerobic fungi and ciliate protozoa are also present. The bacterial communities observed here were similar to those analysed previously, being dominated by Firmicutes and Bacteroidetes [13]. The ruminant faecal microbiota has been examined previously [49,50], with results similar to those presented here where Bacteroidetes are less abundant in faecal samples com-pared to the ruminal samples. It was hoped that by remaining at the order and phyla levels for the bacterial and eukaryotic analysis that we would be able to observe some level of corresponding microbial presence that could be attributed back from the faecal samples to the ruminal samples. Whilst they held a general similarity the microbiomes varied between sample sites, there was obvious differences in the relative abundancies. This is most likely due to the change in roles from fore-gut to hind-gut. It has been pre-viously suggested that whilst the rumen is responsible for the decomposition of plant cell materials [51]. This is an essential step in the absorption of nutrients in cattle, where higher cellulolytic activity can lead to higher representations of Bacteroidetes and Spirochaetes. This differs from the hind-gut, where the introduction of fluid and bile aid the intestines in breaking fat, protein, and starch leading to whereas higher proteolytic activity present in the hind gut can lead to the dominance of Firmicutes and Proteobacteria [49,52]. The ratio of Bacteroidetes to Firmicutes has been seen to alter due to changes in feed [53]. Faecal sampling could be a non-invasive way of attempting to link changes in bacterial abundancies to further functional traits. Furthermore, from a breeding perspective the observation of core species through faecal sampling could enable hologenomic selection approaches to be utilised in cattle [23]. As such the use of faecal samples to generate microbial similarity matrices may be more effective that from oral samples due to the need for a high level of reproducibility and repeatability. For oral sampling, this may pose difficulties due to the need to capture samples close to rumination for accurate results [16]. This is not an issue that faecal sampling faces. This is, however, beyond the scope of this study and would have to be proven in a large number of animals..”

  1. This was deliberately selected, as the individual variations were not of interest to us in this study, we were more interested in what is the impact of the sample site and extraction method and how these effected the observable data.

  1. All specific comments have been addressed and can be seen in the manuscript.

Round 2

Reviewer 2 Report

The authors responded appropriately to my comments, modifying the paper accordingly.

Nonetheless,, I still have one comment: Table 2 showed a p value <0.03. Usually in this case a p value <0.05  is used. I don't remember ever seeing p <0.03 reported. It is therefore suggested to modify it accordingly.

Author Response

Dear reviewer,

Thank you for taking the time to again review the article and for your comment:

“Table 2 showed a p value <0.03. Usually in this case a p value <0.05  is used. I don't remember ever seeing p <0.03 reported. It is therefore suggested to modify it accordingly.”

We have modified the p value in table 2 from p <0.03 to p <0.05, and hope that this adaptation to the manuscript is adequate for your approval.

Reviewer 3 Report

Dear Authors. I am satisfied with the correction. 

Author Response

Dear reviewer,

Thank you for taking the time to again review the article and we hope that these adaptations to the manuscript are adequate for your approval.